# Performance Improvement of Single-Frequency CW Laser Using a Temperature Controller Based on Machine Learning

**DOI:** 10.3390/mi13071047

**Published:** 2022-06-30

**Authors:** Haoming Qiao, Weina Peng, Pixian Jin, Jing Su, Huadong Lu

**Affiliations:** 1State Key Laboratory of Quantum Optics and Quantum Optics Devices, Institute of Opto-Electronics, Shanxi University, Taiyuan 030006, China; 202022616076@email.sxu.edu.cn (H.Q.); 201912607006@email.sxu.edu.cn (W.P.); pxjin@sxu.edu.cn (P.J.); jingsu@sxu.edu.cn (J.S.); 2Collaborative Innovation Center of Extreme Optics, Shanxi University, Taiyuan 030006, China

**Keywords:** stable single-frequency laser, temperature control, machine learning, BP neural network, PID control

## Abstract

The performance improvement of an all-solid-state single-frequency continuous-wave (CW) laser with high output power is presented in this paper, which is implemented by employing a temperature control system based on machine learning to control the temperature of laser elements including gain crystal, laser diode and so on. Because the developed temperature controller based on machine learning combines the back propagation (BP) neural network algorithm with the proportion-integration-differentiation (PID) control algorithm, the parameters of the PID are adaptive with the variation of the environment. As a result, the control speeds and control abilities of the temperatures of the elements are dramatically enhanced. In this case, the output characteristic and the adaptability to the environment as well as the stability of the single-frequency CW laser are also improved greatly.

## 1. Introduction

All-solid-state continuous wave (CW) single-frequency lasers have been applied in quantum optics and quantum information [1,2], precision measurement [3], optical holography [4], optical storage [5], cutting [6], welding [7,8,9], sensing [10,11] and so on, owing to their intrinsic advantages of compact structure, high stability, low intensity noise and high beam quality [12,13]; however, in order to attain a stable all-solid-state single-frequency CW laser with high output power, the temperatures of the pump source and gain crystal as well as nonlinear crystal must be precisely controlled in addition to the design of a unidirectional resonator to eliminate the spatial hole burning effect [14]. Especially for the gain crystal, in the process of laser emission, a lot of waste heat is generated due to quantum defect, energy transfer upconversion (ETU), excited state absorption (ESA) and cross relaxation (CR), and dissipates within the host lattice, which can change the operating temperature of the gain crystal and further induce the thermal lens effect, thermal astigmatism and so on [15,16]. In other words, it is difficult to obtain a stable single-frequency CW laser if the temperature of gain crystal cannot be controlled well—the gain crystal may be damaged; therefore, it is necessary to accurately control the temperature of the optical elements by designing and building a temperature controller with good performance. Traditional proportion-integration-differentiation (PID) control is one of the earliest developed control strategies, which can meet various control requirements for different controlled objects because of the unique PID control algorithm [17]. The basic principle of traditional PID control is to create a closed control loop to reduce the error between the measured and set values. Despite its robustness and reliability, traditional PID control suffers from a major drawback of fixed PID parameters, which directly influences its control abilities [18]. In general, an empirical method is widely used in many control systems to optimize PID parameters, which can not only lose a lot of time but also adapt badly to different environments.

Emerging technologies such as artificial intelligence and cloud computing have been widely used in the field of industrial production, yielding positive results. Machine learning is a new research field of artificial intelligence. Using machine learning, a robot or computer can automatically obtain action parameters, which paves a good way to auto-optimize the PID parameters with the variation of the environment after combining traditional PID with machine learning. As early as 1943, W. S. McCulloch and W. Pitts proposed a theory of artificial neural network (M-P model) according to the structure and working mechanism of biological neurons, which has become a cornerstone of the neural network [19]; however, the weight value in the M-P model was fixed. In order to adjust the weight value to achieve the optimal value and expand the neurons number of output layer, the back propagation (BP) neural network was proposed by D. Rumelhart in 1986 [20]; however, limited by the computational power and calculation speed [21] of the computer, the BP neural network remains in the theoretical stage. Around 2010, benefiting from improvement of the computation speed and the emergence of the big data, all types of machine learning algorithms including the BP neural network achieved remarkable results in the fields of artificial intelligence such as computer vision, intelligent speech recognition and autonomous driving [22,23,24,25]. Especially, Y. Zhou et al. compared the performance of reactor before and after combining the BP neural network with the PID temperature control system in 2009 [26]. The simulated results revealed that the dynamic response speed of the system was dramatically improved and the system overshoot was effectively reduced when the PID temperature control system was based on the BP neural network. In this paper, we present an all-solid-state high-power single-frequency continuous-wave (CW) laser with good performance, which is implemented by employing a temperature control system based on machine learning.

## 2. Experiment Design and Theory

### 2.1. Experiment Setup

The performance improvement experiment was implemented in a homemade high-power single-frequency CW 532 nm laser, which is depicted in Figure 1. In order to achieve high output power by effectively reducing the thermal effects of the gain crystal, a fiber-coupled laser diode (LD) with the center wavelength of 888 nm acted as the pump source [27]. Its fiber core diameter, numerical aperture and maximum output power were 400 µm, 0.22, and 110 W, respectively. A telescope coupling system consisted of two lenses with focal lengths $f_{1}$ = 30 mm and $f_{2}$ = 80 mm, respectively, was employed to optimize the beam size of the pump laser to achieve the optimal mode-matching of the laser.

The resonator of the laser was a butterfly-shaped ring cavity including 4 mirrors. The input coupling mirror M${}_{1}$ was a concave–convex lens with the curvature of 1500 mm, which was coated with high-transmission (HT) film at 888 nm (T${}_{888\mathrm{nm}}$ > 99.5%) and high-reflection (HR) film at 1064 nm (R${}_{1064\mathrm{nm}}$ > 99.7%), respectively. M${}_{2}$ was a plane-convex mirror (R = 1500 mm) coated with a high-reflection (HR) film at 1064 nm (R${}_{1064\mathrm{nm}}$ > 99.7%). M${}_{3}$ and M${}_{4}$ were both plane—concave mirrors with the curvature radius of 100 mm (R = −100 mm), where M${}_{3}$ was coated with HR film at 1064 nm (R${}_{1064\mathrm{nm}}$ > 99.7%), and the output coupler M${}_{4}$ was coated with partial transmission (T${}_{1064\mathrm{nm}}$ = 1.5%) at 1064 nm and HT film (T${}_{532\mathrm{nm}}$ > 95%) at 532 nm [28]. A type I non-critical phase-matched lithium triborate (LBO) crystal was inserted into the cavity for intracavity frequency-doubling. To ensure the unidirectional operation of the laser, an optical diode (OD) including a terbium gallium garnet (TGG) crystal surrounded by a magnetic field and a half-wave plate (HWP) was adopted.

### 2.2. Design of Laser Temperature Control System

The designed and built temperature controller of the all-solid-state continuous wave (CW) single-frequency laser contained four modules [29], which were utilized to control the temperatures of laser diode, gain crystal, LBO crystal and the resonator, respectively.

The temperature control process of the every module was divided into five parts, which include temperature acquisition unit, micro control unit (MCU), temperature control unit, controlled object and display panel. The thermistor TCS610 was used as a temperature acquisition unit considering its characteristics of low cost, small size and wide measurement range. A digital signal processor (DSP) chip (TMS320F28069, Texas Instruments, Dallas, TX, USA) was employed as the MCU owing to its excellent processing speed. The motor driver (DRV8432, Texas Instruments, Dallas, TX, USA) and the thermoelectric cooler (TEC, Ferrotec, Hangzhou, China) were utilized as the temperature control unit to heat or cool the target objects by changing the supplied current. In addition, the display panel was connected to temperature collection device and MCU by RS232 serial port communication. The specific system was shown in Figure 2.

The temperature information of the controlled object was firstly collected by the thermistor TCS610 and its corresponding temperature value was displayed on the host computer screen through the RS232 serial port. At the same time, this value was sent to the MCU for BP neural network PID calculation. A pulse width modulation (PWM) signal was then obtained after the calculation, which was further transmitted to the motor driver (DRV8432) to ensure the supplied driver current to TEC according to the duty cycle of PWM signal. Once the driver current was loaded to the TEC, the controlled object was heated or cooled since the controlled object and TEC were tightly attached together by thermally conductive silicone grease. In this case, the temperature of the controlled object would vary towards the target temperature value. At this moment, a closed-loop control system was built and a whole control cycle was spent in 0.0875 s; however, in the actual process, several closed-loop control cycles were needed to make sure that the temperature of the controlled object was stably controlled at the set point by the BP neural network PID controller. It can also be seen that the BP neural network PID was the most crucial part of the whole closed-loop control system from the framework shown in Figure 2 because the calculated results directly influenced the driver currents loaded to TEC.

### 2.3. Theory of BP Neural Network PID

The machine learning algorithm of BP neural network was utilized to real-time adjust PID parameters according to the operating state of the laser [30]. By referencing the theory of the BP neural network [20] and the online learning strategy [31], online learning BP neural network algorithm was designed in temperature controller to enhance the self-adaptation and improve the performance of the all-solid-state CW single-frequency laser, as shown in Figure 3. The BP neural network consisted of three different sub networks: input layer, hidden layer, and output layer. In this network, there were three neurons in the input layer, which were the amount of error: Δe(k), the trend of error: e(k), and the change of error: Δe(k)−Δe(k−1). The output layer also has three neurons, corresponding to the proportional coefficient: $K_{p}$, the integral coefficient: $K_{i}$, and the derivative coefficient: $K_{d}$. Considering the finite computational power and calculation speed of MCU, a relatively simple and classical BP neural network algorithm with one hidden layer was employed. Based on empirical formulas: $\sqrt{m+n}+a$, the limited number of the hidden layer’s neurons was calculated. Where *m* was the number of neurons in the input layer, *n* was the number of neurons in the output layer, and *a* was a constant between 1 and 10. After several tests, the number of hidden layer neurons was defined as 8. In the following formulas, neurons in the input, hidden and output layers were represented by i(1∼3), j(1∼8), and l(1∼3), respectively.

Thus, the input layer neuron was set,
(1)$$\left\{\begin{array}{c} X_{1}=\Delta e\left(k\right)\\ X_{2}=e\left(k\right)\\ X_{3}=\Delta e\left(k\right)-\Delta e\left(k-1\right) \end{array}\right.$$
and the output layer neuron was set,
(2)$$\left\{\begin{array}{c} Y_{1}=\frac{e^{net_{21}}}{e^{net_{21}}+e^{-net_{21}}}=K_{p}\\ Y_{2}=\frac{e^{net_{22}}}{e^{net_{22}}+e^{-net_{22}}}=K_{i}\\ Y_{3}=\frac{e^{net_{23}}}{e^{net_{23}}+e^{-net_{23}}}=K_{d} \end{array}\right.$$

Because the output range of the hyperbolic tangent function shown in Equation (Equation 3) was between −1 and 1, it can be used as an activation function to coordinate the temperature rise or fall, especially in the calculation of hidden layer neuron.
(3)$$tanh\left(j\right)=\frac{e^{j}-e^{-j}}{e^{j}+e^{-j}}$$

However, as shown in Equation (Equation 4), in calculation of output layer neuron, we chose the sigmoid function with a range of 0 to 1 as activation function to coordinate the PID parameters due to the PID parameters need to be positive.
(4)$$sigmoid\left(l\right)=\frac{e^{l}}{e^{l}+e^{-l}}$$

The other formulas about BP neural network were referred to Reference [20].

### 2.4. Design of Parameter Optimizing System

It can be seen from Equation (Equation 2) that all PID parameters calculated by BP neural network were in the ranges of 0 to 1. The obtained parameters cannot be directly used in the temperature controller, owing to the calculated increment of current with these parameters being too small to drive the TEC to effectively and rapidly control the temperature of the crystal. To this end, a parameter optimizing system was designed for temperature controller to further optimize the PID parameters.

In order to make the PID parameters suitable for the temperature controller, the obtained parameters from BP neural network should be amplified by a certain proportion and the amplification factors should be regulated dynamically. Here, the amplification factor of $10^{n}$ was chosen as shown in Equation (Equation 5) and the positive integer *n* was limited within 3, owing to that the calculated control value with n≥4 exceeded the limitation of the control system. Consequently, three groups magnified PID parameters, $K_{p(n)}$, $K_{i(n)}$, and $K_{d(n)}$, were obtained and a group of appropriate parameters could be selected from these three groups parameters in different conditions.
(5)$$\left\{\begin{array}{c} K_{p(n)}=10^{n}\cdot K_{p}\\ K_{i(n)}=10^{n}\cdot K_{i}\\ K_{d(n)}=10^{n}\cdot K_{d} \end{array}\right.$$

For the purpose of selecting the appropriate parameters, the control value was calculated with *n* = 3, 2, and 1 in turn according to Equation (Equation 6). Then, the data were predicted through calculating the increment of current by each group of the amplified PID parameters, which was represented as Equation (Equation 6). The purpose of this step was to prepare for filtering data.
(6)$$\left\{\begin{array}{c} \Delta u_{\left(n\right)}\left(k\right)=K_{p(n)}\left[e\left(k\right)-e\left(k-1\right)\right]+K_{i(n)}e\left(k\right)+K_{d(n)}\left[\Delta e\left(k\right)-\Delta e\left(k-1\right)\right]\\ u_{\left(n\right)}\left(k\right)=u_{\left(n\right)}\left(k-1\right)+\Delta u_{\left(n\right)}\left(k-1\right) \end{array}\right.$$

Lastly, screening out the increment of current, which conforms to Equation (Equation 7). The reason why the maximum increment of current was setup 200 is that the maximum duty cycle of the PWM was 200 in TMS320F28069 chip. The group of the amplified PID parameters with the larger *n* was chosen if not just one increment of current would conform to Equation (Equation 7).
(7)$$0\leq |u_{\left(n\right)}\left(k\right)|\leq 200$$

## 3. Experimental Results

For a high power single-frequency CW laser, the temperature stability of the gain crystal was the most important factor for the stable operation since lots of thermal generated in the process of laser emission could increase the temperature of the gain crystal and then directly restrict the output power and optical conversion efficiency as well as the stabilities; therefore, in the experiment, we paid main attention to the influence of the temperature control ability of the gain crystal on the performance of the single-frequency CW green laser. The output power of the built single-frequency CW 532 nm laser with the increase in the incident pump power was firstly recorded when the traditional PID and BP neural network PID was adopted, respectively, and the results are depicted in Figure 4. In the experiment, increasing the incident pump power can be divided into three stages. In the first stage, the incident pump power automatically increased at a rate of 0.40 W/s before the threshold pump power (37.42 W). Although there was no laser emission, the laser resonator would gradually access the stability range because of the cumulative thermal lensing effect of the gain crystal. Once the incident pump power was beyond the threshold value, a minute was needed to ensure that temperatures of all elements including gain crystal, LBO crystal and the resonator were stabilized to their set values. Thereafter, the incident pump power quickly increased from the threshold value to the optimal operating point (78.00 W) at a rate of 10.15 W/s, and the trend is shown in curve (a) in Figure 4. Curves (b) and (c) are the output power of the achieved laser for the traditional PID and BP neural network PID, respectively. It can be seen that the maximum output power of the 532 nm green laser was about 18 W; however, after the temperature controller based on the traditional PID was replaced by that of the BP neural network PID, the spent time of the process that the output power reached its maximum value from zero was shortened from about 740 s to 240 s. The control speed was increased by 67.6%. Especially, when the traditional PID temperature controller was used in the experiment, serious power fluctuation was observed before reaching the maximum output power of the laser. Even the output power fell down to 6.99 W at 116 s; however, when the BP neural network PID temperature controller was adopted, we only saw the tiny ripples and the output power of the single-frequency 532 nm green laser reached the maximal value fast. Moreover, the output power only dropped to 10.74 W before it started to return. The results proved that the BP neural network PID temperature controller not only had an excellent ability for temperature control of the laser but also greatly enhanced laser output performance.

In the process of recording output power of the achieved laser, the temperature of gain crystal (Nd:YVO${}_{4}$) was also monitored in real time, which is depicted in Figure 5. Curve (a) is still the incident pump power trend, the same as in Figure 4. Curves (b) and (c) are the crystal temperature of the achieved laser for the traditional PID and BP neural network PID, respectively. The set temperature value of gain crystal was 27.10 °C in the experiment. After the incident pump power exceeded the threshold point (37.42 W), the temperature of the gain crystal increased monotonously with the incident pump power until it reached the optimal operating point (78.00 W). Then, in the process of that incident pump power being maintained at the optimal operating point, the temperature of the gain crystal dropped and oscillated several times until it was controlled back to the optimum operating temperature. The time spent in this process was shortened from about 400 s to 200 s. The control speed was increased by 50%. Especially, when the temperature controller was using traditional PID, the temperature fluctuation of the gain crystal was observed—the maximum and minimum values reached up to 29.20 °C and 26.44 °C, respectively; however, when the temperature controller used the BP neural network PID, the temperature of the gain crystal only reached 28.40 °C, and the temperature fluctuation was relatively stable. The control overshoot was decreased by 53%. The results further proved that the BP neural network PID temperature controller can help the gain crystal to control its temperature within the optimal operating temperature range, and further improve the performance of the single-frequency CW green laser.

After the output power of the laser was stabilized, the temperature control abilities of the traditional PID and the BP neural network PID were further compared by decreasing and increasing the temperature of the gain crystal by 1 °C in the experiment, respectively. The results are shown in curves (a) and (b) of Figure 6. When the set temperature of the gain crystal was decreased from 27.10 °C to 26.10 °C, it was found that the spent time of the temperature stably reaching the set value and the maximal overshoot were about 300 s and about 0.1 °C, respectively, when the traditional PID temperature controller was used. Moreover, the multiple temperature oscillations were easily observed in the process of the temperature control; however, once the BP neural network PID temperature controller was employed, this time and overshoot were shortened to 50 s, and near zero, respectively. Further, only several tiny ripples can be observed in the process of the temperature control. The phenomenon for increasing the set temperature value resembled that of the decreasing process. The results revealed that the temperature control speed was significantly improved by replacing the traditional PID temperature controller with the BP neural network PID temperature controller, which was of great significance for attaining a stable single-frequency CW green laser with high output power.

When the temperature of the LBO crystal was controlled to the optimal phase-matching temperature of 147.46 °C, the power fluctuations of the 532 nm laser were measured in 2 h with both temperature controllers, which are shown in Figure 7. When the temperature of the crystal was controlled by the traditional PID controller, the maximum output power was limited to 18.01 ± 0.09 W, and the stability of the output power of the 532 nm laser was ±0.51% in 2 h; however, when the BP neural network PID controller began to work, the maximum output power reached up to 18.03 ± 0.06 W, and the power fluctuation of the 532 nm laser was reduced to ±0.36% in 2 h. The obtained power fluctuation of ±0.36% was less than that of the 532 nm laser controlled by traditional PID controller. The result showed that the presented method of the BP neural network PID control in this paper can also enhance the stability of the 532 nm laser. The comparison of the experimental results between traditional PID and BP-PID is summarized in Table 1.

## 4. Conclusions

In conclusion, we have manufactured a new stable single-frequency CW laser temperature controller based on machine learning, which combined the BP neural network algorithm and PID control algorithm. A BP neural network with PID parameters as output layer neurons has been built, which had the adaptive ability of forward learning and backward training to calculate the most appropriate PID parameters according to different working environments. In the BP neural network, the error, the change of error and the trend of error between the measured and set values were defined as the neuron of the input layer. We used the TMS320F28069, a DSP chip produced by Texas Instruments, as MCU to achieve the calculation of the BP neural network and temperature control of the gain crystal (Nd:YVO${}_{4}$). The performance improvement of single-frequency CW laser in our experiments was achieved by replacing the traditional PID temperature controller with the BP neural network PID temperature controller. The experiment results showed that the speed of that the output power reached the maximum value was increased by 67.6% when the incident pump power was injected from the threshold point (37.42 W) to the optimal operating point (78.00 W) at a rate of 10.145 W/s. Simultaneously, the temperature control overshoot of the gain crystal (Nd:YVO${}_{4}$) was decreased by 53%. Moreover, the power fluctuation of the homemade single-frequency CW 532 nm laser was reduced to ±0.36% in 2 h when the BP neural network PID temperature controller was used. The research illustrated that not only the laser crystal temperature control ability but also the output power stability of the single-frequency CW 532 nm laser can be improved by using the BP neural network PID temperature controller. For the purpose of further improving the laser performance, more complex machine learning algorithms based on the deep neural networks, such as long short-term memory (LSTM), reinforcement learning and so on, processed in the GPU or FPGA will be adopted in the laser temperature control systems to stably control the temperature of the key optical elements in laser resonator and synchronously optimize the laser beam quality, laser linewidth, stability of the single-longitudinal-mode as well as intensity noise. The performance improvement of single-frequency CW laser was of great significance to the study of quantum optics, laser medical technology and laser holography technology.

## Figures and Tables

**Figure 1 micromachines-13-01047-f001:**
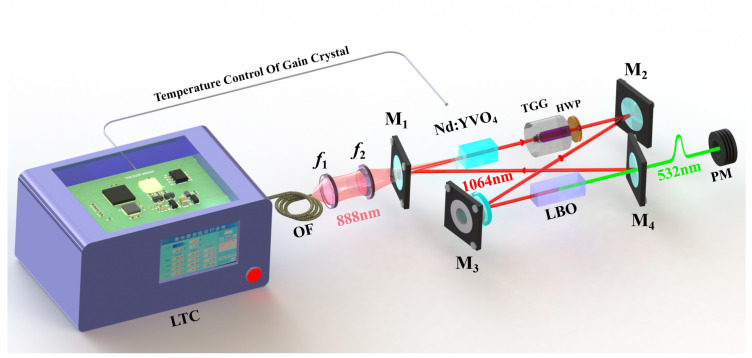
Schematic diagram of high-power single-frequency CW 532 nm laser. LTC, Laser Temperature Controller (Including pump source LD and laser temperature control system); OF, Optical Fibers; $f_{1}$–$f_{2}$, lenses; M${}_{1}$–M${}_{4}$, mirrors; Nd:YVO${}_{4}$, Nd${}^{3+}$ doped yttrium vanadate; HWP, Half Wave Plate; TGG, Terbium Gallium Garnet; LBO, lithium triborate; PM, Power Meter.

**Figure 2 micromachines-13-01047-f002:**
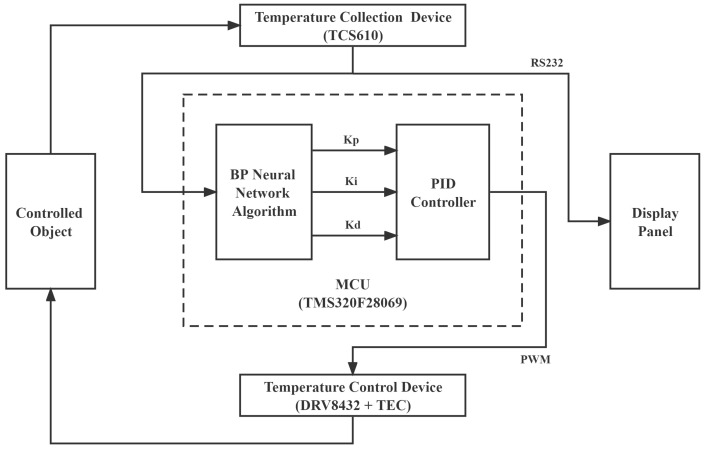
Framework of the Laser Temperature Control System. BP, back propagation; MCU, micro control unit; PWM, pulse width modulation; TEC, thermoelectric cooler; PID, proportion-integration-differentiation.

**Figure 3 micromachines-13-01047-f003:**
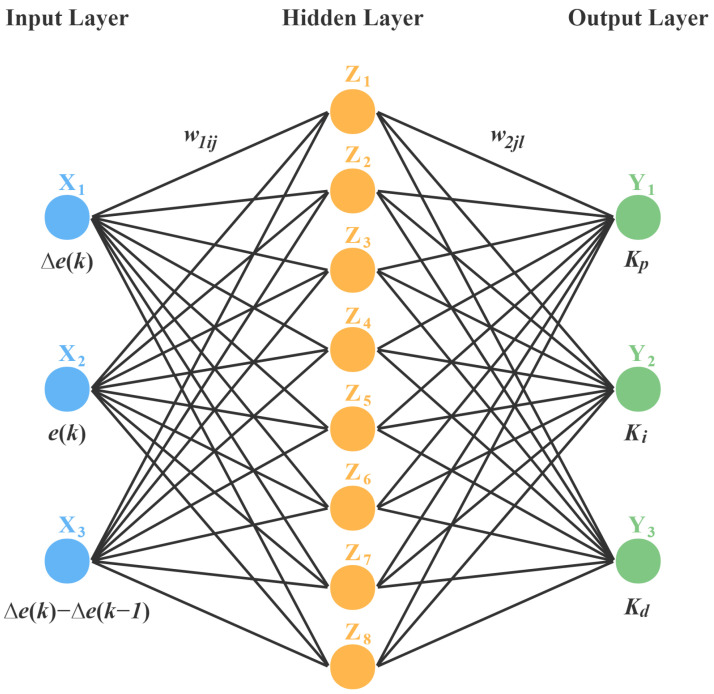
Structure of BP Neural Network.

**Figure 4 micromachines-13-01047-f004:**
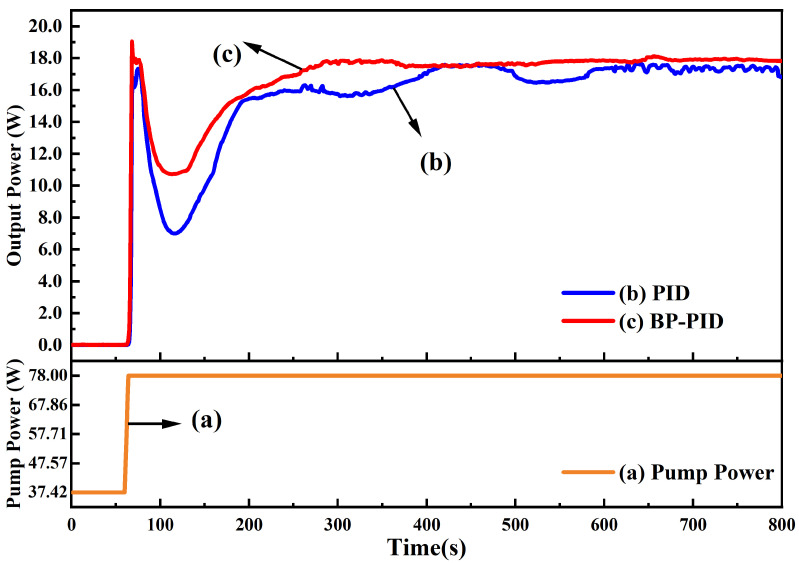
The change of the laser output power when the incident pump power increased from the threshold point to the optimal operating point. The orange line (a) was the change of the incident pump power, the blue line (b) was the laser output power of using traditional PID controller, and the red line (c) was the laser output power using BP neural network PID controller.

**Figure 5 micromachines-13-01047-f005:**
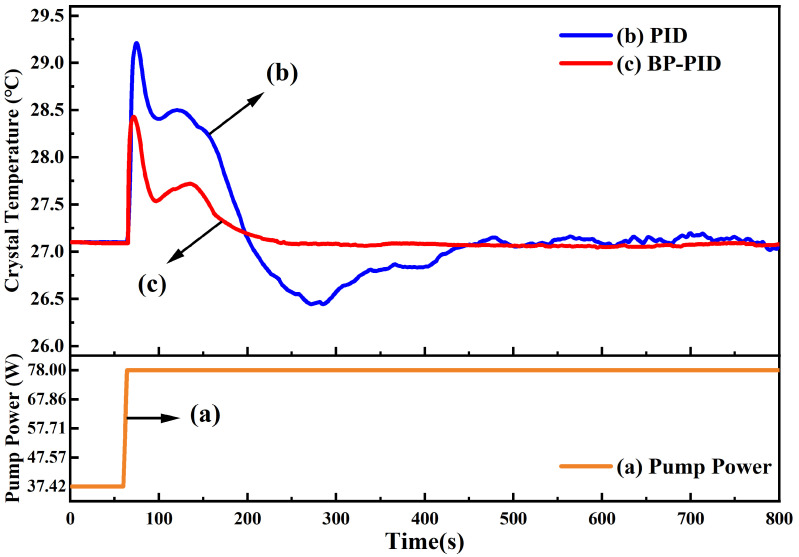
The change of the crystal temperature when the incident pump power increased from the threshold point to the optimal operating point. The orange line (a) was the change of the incident pump power, the blue line (b) was the crystal temperature of using traditional PID controller, and the red line (c) was the crystal temperature of using BP neural network PID controller.

**Figure 6 micromachines-13-01047-f006:**
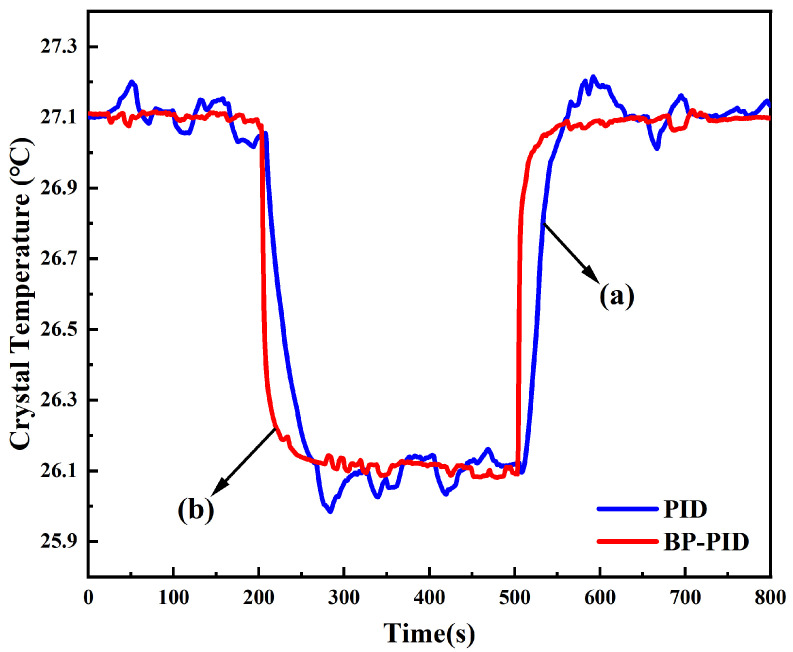
Comparison between traditional PID control and BP neural network PID control when changing the same temperature. The blue line (a) is traditional PID control, and the red line (b) is BP neural network PID control. The temperature decreased by 1 °C in 200∼500 s and increased by 1 °C in 500∼800 s.

**Figure 7 micromachines-13-01047-f007:**
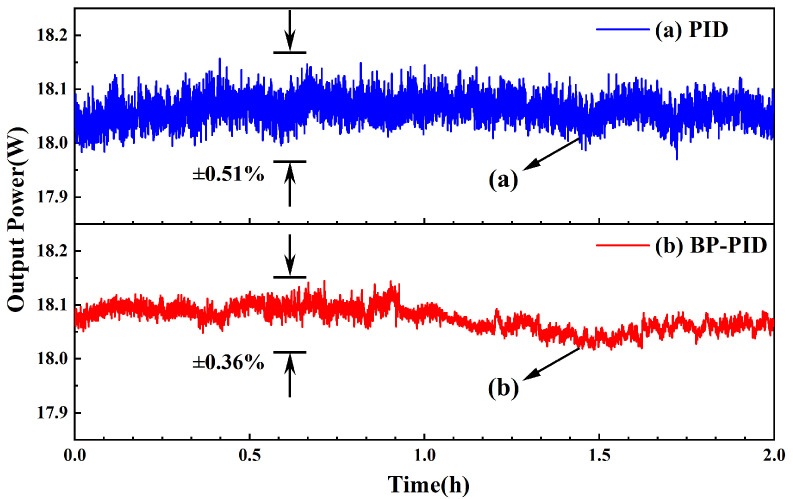
Comparison of output power and its stability between traditional PID control and BP neural network PID control. The blue line (a) is traditional PID control and the red line (b) is BP neural network PID control. Test time was 2 h.

**Table 1 micromachines-13-01047-t001:** Comparison of the experimental results between traditional PID and BP-PID.

Experiment	Increasing the Pump Power to Optimal Operation Point					
	Variation of the Laser Power			Variation of the Crystal Temperature		
	Maximum	Minimum	Time	Maximum	Minimum	Time
**PID**	17.65 W	6.99 W	740 s	29.20 °C	26.44 °C	400 s
**BP-PID**	19.04 W	10.74 W	240 s	28.40 °C	27.10 °C	200 s
**Experiment**	**Changing the Crystal Temperature Set Value**				**Stability of the Output Power (2 h)**	
	**Decrease (−1 °C)**		**Increase (+1 °C)**			
	**Overshoot**	**Time**	**Overshoot**	**Time**		
**PID**	0.1 °C	>300 s	0.1 °C	>300 s	±0.51%	
**BP-PID**	<0.01 °C	<50 s	<0.01 °C	<50 s	±0.36%	

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
