# Peer review of "Performance Improvement of Single-Frequency CW Laser Using a Temperature Controller Based on Machine Learning"

_micromachines, 2022, doi:10.3390/mi13071047_

Round 1

Reviewer 1 Report

Whilst the results are interesting and clear, several comments need to be addressed before publication. In addition, the number of references needs to be increased to accommodate the necessary literature and aid in the reader’s understanding. Furthermore, the grammar needs improvement as highlighted below.

1.       It would be useful to the reader why only one hidden layer neural network? When other machine learning networks use much deeper networks.

2.       What was the shortest resolution time of temperature/power control?

3.       Additional references in the introduction could be added to make it more complete, such as for cutting, welding and sensing:

a.        Ghany, Khalid Abd El and Mohamed Newishy. “Cutting of 1.2 mm thick austenitic stainless steel sheet using pulsed and CW Nd:YAG laser.” Journal of Materials Processing Technology 168 (2005): 438-447.

b.       Suder, W. J., and S. Williams. "Power factor model for selection of welding parameters in CW laser welding." Optics & Laser Technology 56 (2014): 223-229.

c.        Grant-Jacob, James A., et al. "Particle and salinity sensing for the marine environment via deep learning using a Raspberry Pi." Environmental Research Communications 1.3 (2019): 035001.

d.       Larsson, Jim, Joakim Bood, Can Xu, Xiong Yang, Robert Lindberg, Fredrik Laurell and Mikkel Brydegaard. “Atmospheric CO2 sensing using Scheimpflug-lidar based on a 1.57-µm fiber source.” Optics express 27 12 (2019): 17348-17358 .

A useful reference for the evolution of graphics cards could be added to the introduction on machine learning (line 52):

e.        Evolution of the Graphics Processing Unit GPU) Dally W, Keckler S, Kirk D IEEE Micro (2021) 41(6) 42-51

Regarding BP neural networks, other laser-based references could be useful to include:

f.        Zhang, Yanxi, Xiangdong Gao, and Seiji Katayama. "Weld appearance prediction with BP neural network improved by genetic algorithm during disk laser welding." Journal of Manufacturing Systems 34 (2015): 53-59.

g.       Wu, Bing, et al. "Error compensation based on BP neural network for airborne laser ranging." Optik 127.8 (2016): 4083-4088.

4.       In the latter stages of the manuscript, I think it would be of benefit to the readers if the authors could add 1) how to further improve the results, and 2) what the next steps might be. For example, for controlling a laser system that is more complex, additional types of neural networks may need to be considered, such as reinforcement learning. Please see following references:

h.       Masinelli, Giulio, et al. "Adaptive laser welding control: A reinforcement learning approach." Ieee Access 8 (2020): 103803-103814.

i.         Degrave, Jonas, et al. Magnetic control of tokamak plasmas through deep reinforcement learning Nature (2022) 602(7897) 414-419

5.       Regarding figures:

Figure 1: The laser direction arrows could be more define and the yellow line of temperature controller is hard to see.

Figure 2: State what MCU, TEC and PID is in the caption.

6.       Some grammatical checking needs to be done. See for example some suggestions below:

Line 22: “a lot of waste heat generate”, should be “a lot of waste heat is generated”

Line 27: “control”, replace with “controlled”.

Line 28: “well”, replace with “accurately” as this would be a more suitable word.

Line 33: “creating”, replace with “is to create”

Line 39:  Reword sentence.

Line 41: “By machine learning, robot or computer”, replace with “Using machine learning, a robot or computer”

Line 66: “was”, replace with “is”

Line 148: “was”, replace with “being”

Line 197: “166th”, replace with just 116”

Line 231: “300 seconds about 0.1°C”, replace with “300 seconds and about 0.1°C”

Line 256: “BP neural network”, should be “A BP neural network”.

Reviewer 2 Report

The authors attempt to improve the performance of single-frequency CW laser using a temperature controller based on machine learning. 

1. First, the experimental design for machine learning experiments is not clear.  There is no description of a training and test-set, or of cross-validation. So it is not clear if the method proposed by the authors is over-fitting to the problem or not.  The number of data-points in the datasets are also not clearly mentioned. It is important for the dataset to be divided into train and test, and the model (trained on training) results to be tested on the hold-out test set. At the very least, there should be a cross-validation model where out of fold predictions are compared with that of the traditional PID based results. 

2. It is unclear why neural network was chosen instead of other alternatives such as a generalized linear model. A GLM is much more explainable compared to a neural network given that a 1-hidden layered neural network is not very complex. 

3. It is unclear if the authors tried using other features in their feature set or this was the only set of features used. Further, it is unclear why a 1-hidden layer model is used instead of more hidden layers. 

4. The authors should provide summary of comparison between traditional PID and BP-PID results in a table so that it is easy to compare the results. 

Round 2

Reviewer 2 Report

The authors have provided sufficient response to first round of comments.